# Spatial and temporal differences and convergence analysis of multidimensional relative poverty in ethnic areas

**Jing Cheng, Xiaobin Yu**[ORCID]*

Jiangsu University of Science and Technology, School of Humanity & Social Science, Zhenjiang, China

* 231111201104@stu.just.edu.cn

**Data Availability Statement:** All data are in the manuscript and supporting information files.

**Funding:** This study was funded by: National Social Science Fund of China (Grant numbers 23BJY182); The Ministry of education of Humanities and Social

## Abstract

Reducing multidimensional relative poverty is one of the important issues in the current global poverty governance field. This article takes 12 ethnic regions in China as the research object and constructs a multidimensional relative poverty measurement system. The calculated multidimensional relative poverty index is decomposed according to provinces, cities, dimensions, and indicators. Then, the Dagum Gini coefficient and convergence analysis are used to analyze spatiotemporal heterogeneity and convergence characteristics. The results show that the multi-dimensional relative poverty situation of various provinces in ethnic minority areas has improved from 2012 to 2021, among which Tibet province is the most serious and Shaanxi is the best. According to the analysis of convergence, it was observed that there is no σ-convergence of multidimensional relative poverty in ethnic areas in general, and there is absolute β-convergence in general and in the southwest and northwest regions, and there is no absolute β-convergence in the northeast region. Based on this, policy recommendations for reducing multidimensional relative poverty are proposed at the end of the article. Compared with previous studies, this article focuses on ethnic regions that are easily overlooked. Starting from the dimensions of economy, social development, and ecological environment, the poverty measurement system has been enriched.

## Introduction

China has eliminated absolute poverty after 2020 and is now shifting towards alleviating relative poverty and achieving common prosperity. For many years, China has attached great importance to poverty management in ethnic areas. However, due to the poor ecological environment, frequent natural disasters, a weak industrial base, and insufficient endogenous motivation for the self-development of poor subjects, ethnic areas have a high degree of vulnerability and a high incidence of relative poverty, making the task of consolidating the effectiveness of poverty eradication all the more urgent and arduous [1]. China has consistently emphasized the need to solidly promote common prosperity, especially in the new stage of development, and has focused on guaranteeing stable poverty eradication and establishing a long-term mechanism for the governance of relative poverty. Therefore, how to consolidate

Science project of China (Grant numbers 22YJA790010). The funders had no role in study design, data collection and analysis, decision to publish, or preparation of the manuscript.

**Competing interests:** The authors have declared that no competing interests exist.

the results of poverty eradication, alleviate relative poverty, enhance the efficiency of relative poverty governance, and realize common prosperity has become an important issue that needs to be resolved urgently. In the long run, eliminating difficulties such as relative poverty, relative backwardness, and regional disparities will be a major challenge. To solve the problem of relative poverty, the first thing to do is to develop accurate methods for identifying relative poverty and construct a reasonable multidimensional indicator system for relative poverty. In addition, as the indicator system changes, it will inevitably have an impact on identifying impoverished groups, so dynamic identification methods should be adopted to consolidate poverty alleviation achievements. Among these, relative poverty refers to the lack of opportunities, capabilities, or rights, and the conditions and opportunities in terms of food, housing, and entertainment are lower than the average level of the society, which essentially belongs to the category of multi-dimensional poverty, including both the economic dimension reflecting the "poverty" and the social and development dimension reflecting the "hardship" [2]. Multidimensional poverty implies deprivation of capabilities, rights, and well-being. It includes not only income poverty but also types such as health poverty and ecological poverty, which are specific manifestations of relative poverty [3]. As a national underdeveloped region, solving the problem of relative poverty in ethnic areas will not only benefit the local poor but also drive the development of the whole ethnic area, generating a large number of "spillover effects", which will have a demonstration effect on the promotion of high-quality development in ethnic areas and the realization of common prosperity and development of all ethnic groups. Therefore, an examination of the current status of the development of multidimensional relative poverty in ethnic areas and the characteristics of the differences will systematically present the spatial and temporal changes in multidimensional relative poverty in ethnic areas and the characteristics of the regional differences, which will help to provide policy references for controlling the risk of returning to poverty and formulating long-term anti-return-to-poverty policies in the context of comprehensive poverty eradication.

From the existing studies, scholars have made theoretical analyses of the connotation and performance characteristics of multidimensional relative poverty and made empirical studies on the construction and measurement of the indicator system. The concept of multidimensional relative poverty was first put forward by the World Bank in 2003, and the core idea is that human poverty is not only embodied in the lack of income, but also includes other objective indicators, such as the lack of drinking water, roads, sanitation facilities, and other aspects, as well as the subjective perception of the dimension of poverty [4]. Subsequent scholars have expanded the definition of relative poverty and absolute poverty to continuously enrich the theoretical study of multidimensional relative poverty. In analyzing poverty in an aging society, Weihong Zeng defines relative poverty as the relative deprivation of an individual in a certain situation versus the highest situation [5]. Compared with multidimensional poverty and relative poverty, multidimensional relative poverty reflects the comprehensiveness of multidimensional poverty in measuring poverty from multiple dimensions and highlights the subjectivity of relative poverty in identifying poverty based on the average income level of society [6]. Compared with multidimensional absolute poverty, the key difference is that the criteria for multidimensional relative poverty can be adjusted annually according to economic development and social progress [7]. The flexibility thus achieved makes multidimensional relative poverty more adaptable to social changes and development in terms of identifying and measuring poverty. At the same time, the measurement of multidimensional relative poverty is also more realistic because it can reflect people's poverty more accurately in multiple dimensions. Consequently, multidimensional relative poverty is a dynamic concept, and its connotation and criteria will change over time and space.

At present, the research on multidimensional poverty focuses on the measurement of poverty index and the construction of indicator systems. Initially, poverty was measured singularly from the income dimension [8], and with the development of society, this method is no longer sufficient to truly reflect the poverty situation of a country. Therefore, a multidimensional measurement method was introduced, through which alternative perspectives were provided for the identification of poverty. From the perspective of multidimensional poverty, Yanhui Wang integrates indicators of housing, health, and education into the evaluation system to comprehensively measure and analyze rural poverty [9]. The United Nations Development Program selected 10 indicators from the three domains of education, health, and living standards to calculate the global multidimensional poverty index, and its indicator system has been widely cited by other scholars [10]. On the path of multidimensional poverty research, many scholars have also explored several new research directions including natural shocks [11] and dynamic evolution [12]. Weiming Li selected 10 indicators from the economic dimension and used spatially explicit models to study and analyze the driving factors behind regional income inequality [13]. Nighttime lighting data is an excellent data source for poverty monitoring [14], and its distribution in different regions is severely constrained by regional environment and production factors. A multidimensional poverty indicator system can be constructed from four aspects, namely, industry, education, tourism, and agriculture, using the average nighttime lighting index, the county multidimensional development index, and relative poverty identification methods [15]. In addition to this, new perspectives such as health poverty [16], energy poverty [17], and asset poverty [18] have been introduced into the field. These concepts are not entirely new, but in the past, they were seldom used and measured individually. The introduction of new perspectives makes it easy to identify the type of poverty of an individual and make targeted poverty alleviation measures. Nevertheless, there are some limitations in these approaches, such as the lack of objectivity in selecting the boundaries of cognitive impairment, interpersonal relationships, and social trust, which are mainly based on subjective social experiences; and secondly, the indicators do not cover a wide enough range of topics, which may affect the accuracy of the results. Based on previous studies on multidimensional poverty and the current state of economic and social development, Huanqi Luo also chose four dimensions, which are economic development opportunities, potential development opportunities, internal risks, and external risks, to measure the degree of relative poverty in the region from the perspectives of opportunities and risks [19]. These studies provide a more comprehensive and in-depth reference perspective for us to understand and solve the problem of multidimensional poverty.

In addition to the selection of indicators and the construction of the system, scholars have also paid attention to the study of influencing factors [20], spatial and temporal differences [21], measurement methods [22], and paths to poverty alleviation [23]. The influencing factors of multidimensional relative poverty can be summarized into two parts: one is the intrinsic factors including individual ability, information access, family characteristics, etc.; the other part is the extrinsic factors including social system, geographic distribution, and market structure. Alkire, Center Director of the Oxford Centre for Poverty and Human Development, used panel data to study the dynamic evolution of multidimensional poverty and income poverty and their differences in China and assessed the robustness and stability of multidimensional indicators to changes in weights and indicators [24]. In the global area, he provides projections of multidimensional poverty for 75 countries and also explores the impact of COVID-19 on global levels of multidimensional poverty [25]. Methodological advances are presented to identify the time path of multidimensional poverty on the one hand, and empirical results useful for policy purposes are provided on the other. Specifically for Chinese scholars, Yao Wang reveals whether income poverty is necessarily energy poverty to test the effectiveness of

domestic energy support policies targeting income-poor households, and analyzes the factors that lead to the inconsistency between these two poverty levels [26]. Based on microdata, Xiao-lin Wang measures the multidimensional poverty of urban and rural households in China, arguing that provinces need to strengthen their urban anti-poverty efforts, and that multidimensional relative poverty can be disaggregated by urban and rural areas, regions, age, and gender to adopt more targeted poverty reduction policies [27].

In summary, scholars have conducted extensive research on the field of multidimensional relative poverty. Significant research results have been achieved in both the theoretical analysis of multidimensional poverty and relative poverty, as well as the exploration of poverty influencing factors and policy formulation. However, through in-depth reading of the literature, it has been found that there may be several shortcomings in current research on multidimensional relative poverty. Firstly, most of the current research focuses on the overall country or individual provinces, which makes it difficult to identify poverty differences in different regions and also makes it difficult to see poverty associations between regions. Secondly, in terms of indicator selection, many studies use international standards and the poverty situation of developed countries as benchmarks, which may result in the selected indicators not being suitable for measuring the multidimensional relative poverty level in ethnic regions of China. On the premise of borrowing and learning from the research results of the former, there may be several innovations in this article: firstly, this article focuses on ethnic regions that are easily overlooked. Ethnic regions have complex and diverse geographical, cultural, and economic characteristics, with significant spatial differences in multidimensional poverty. This article uses the AF method to measure the multidimensional relative poverty level in ethnic areas, providing theoretical value for subsequent research on its influencing factors. Secondly, by combining the characteristics of ethnic regions and specific poverty situations, the multidimensional relative poverty measurement system has been enriched. This article selects evaluation indicators that are more in line with the actual situation of ethnic regions, and obtains a more accurate and practical poverty index through the constructed evaluation system. Thirdly, this article uses the Dagum Gini coefficient to analyze the spatiotemporal differences in multidimensional poverty in ethnic regions, taking into account the differences and sources of poverty between different regions.

## Methods

This paper takes 12 provinces in China's ethnic areas as the research object, draws on the three dimensions of economy, social development, and ecological environment proposed by Wang Xiaolin to construct a multidimensional relative poverty system [27], and adopts the A-F method to measure the multidimensional relative poverty index in China's ethnic areas as well as the decomposition of the poverty index by region. Then, the entropy method is used for comprehensive evaluation, and the Gini coefficient is calculated based on the evaluation scores, and divided into three parts, namely, intra-group, inter-group, and hypervariable density, to analyze the spatial and temporal differences. Finally, a convergence analysis is carried out, and governance suggestions are made based on the results.

### Double threshold value method

The A-F method is a widely used poverty measurement method in the international arena for multidimensional poverty. In this essay, we mainly measure from the following three steps: first, we measure whether the poverty level of each province on a certain poverty indicator exceeds a certain deprivation threshold (i.e., the poverty line); next, according to the deprivation thresholds of the 15 poverty indicators determined by the proportion method, we assign

weights to the poverty status on each poverty indicator; finally, according to the weights of the various poverty indicators and the value of the relative poverty, we measure the multidimensional relative poverty index.

Assuming that there are n provinces and d indicator strata, it is then able to obtain the matrix $A_{n \times d}$, and each element $A_{ij}$ in this matrix denotes the value of province i on the jth poverty indicator, where i = 1,2,. . .,n; j = 1,2,. . .,d. First is the one-dimensional identification, taking $Z_j$ to denote the cut-off value (i.e., the poverty line) of the jth poverty indicator, and obtaining the deprivation matrix G = $[G_{ij}]$, based on the matrix A and the cut-off value, where the element $G_{ij}$ is defined as follows: when $A_{ij} < Z_{ij}$, $G_{ij} = 1$; when $A_{ij} \geq Z_{ij}$, $G_{ij} = 0$. For the ijth element, when a province is deprived on the jth poverty indicator, it is assigned a value of 1; and the vice versa is 0. For example, when the critical value of the per capita consumption expenditure of the population is determined to be 10,000 yuan, if the per capita consumption expenditure of a province is 12,000 yuan, at this moment, it exceeds the critical value of deprivation, i.e., $A_{ij} > Z_{ij}$, then $G_{ij} = 0$. Define a column vector $c_i$ to represent the total number of deprivation dimensions endured by individual i. The number of deprivation dimensions is defined as a function of the number of deprivation indicators. Multidimensional identification is then performed, with $f_k$ being the function that identifies poor provinces when k poverty indicators are considered. When $c_i \geq k$, $f_k (a_i;z) = 1$; and vice versa is 0. Simply, when the total number of dimensions of deprivation ($c_i$) that individual i endures is $\geq$ k, $f_k$ defines individual i as a deprived province when the poverty indicator is equal to k; and as a non-deprived province otherwise. After identifying the deprivation of each poverty indicator, the dimensions are summed to obtain a multidimensional composite index. The formula is as follows:

$$M_0 = HA \tag{1}$$

$M_o$ is an adjusted multidimensional poverty index consisting of two components, H (poverty incidence) and A (poverty deprivation share), where H = $q/d$ and A = $|c(k)|/qd$.

The multidimensional relative poverty index is decomposed according to different groups such as indicators, regions, and provinces. Taking district decomposition as an example, assuming that there are p districts in total, where n represents the number of samples in each district, when decomposed according to districts, the overall multidimensional relative poverty index M expression can be written as:

$$M = \sum_{i=1}^{p} \frac{n_i}{n} M_i = \frac{n_1}{n} M_1 + \frac{n_2}{n} M_2 + \cdots\cdots + \frac{n_p}{n} M_p \tag{2}$$

where $M_i$ is the poverty index for the area i. Thus, the contribution of area i to the multidimensional relative poverty index can be obtained as:

$$G_i = \frac{n_i}{n} \frac{M_i}{M} \tag{3}$$

## Dagum Gini coefficient

The Dagum Gini coefficient is used to observe inter-regional differences. The Dagum Gini coefficient decomposes the overall differences into three parts: intra-group differences, inter-group differences, and hypervariable density, to accurately identify the sources of differences, at the same time, it can also solve the problems of small samples and the asymmetry problem. To analyze the regional differences in multidimensional relative poverty in China's ethnic areas, this paper uses the Dagum Gini coefficient and its subgroup decomposition to explore the evolution and sources of inter-regional relative differences. The Gini coefficient G is

calculated as follows:

$$G = \frac{\sum_{j=1}^{k} \sum_{h=1}^{k} \sum_{i=1}^{n_j} \sum_{r=1}^{n_h} |y_{ij} - y_{hr}|}{2n^2 \bar{y}} \qquad (4)$$

In the formula, G is the overall Gini coefficient, k denotes the number of regions divided, and in this paper, k = 5 denotes the five regions divided into South China, North China, Southwest China, Northwest China, and Northeast China, n is the number of provinces, $y_{ij}(y_{hr})$ denotes the comprehensive evaluation scores of the provinces within the region of j(h), and $\bar{y}$ denotes the average value of the multidimensional relative poverty index of the ethnic regions.

According to the Dagum Gini coefficient decomposition method, the overall Gini coefficient is decomposed into three components: intra-regional variance Gw, inter-regional variance Gb, and hypervariable density variance Gt, and G = Gw+Gb+Gt. The intra-regional variance Gw represents the contribution of the difference in the composite score of the multidimensional relative poverty within the five major geographic regions of South China, North China, Southwest China, Northwest China, and Northeast China, and the inter-regional Gb represents the contribution of the difference between the five major regions. Gw denotes the contribution of differences between the five regions, inter-regional Gb denotes differences between the five regions, and hypervariable density Gt denotes the contribution of inter-sample overlap to the overall differences, which in this study mainly refers to the increase in overall differences triggered by the emergence of high-poverty regions within the multidimensional relative low-poverty group, and then the emergence of low-poverty regions within the multidimensional relative high-poverty group.

## Convergence analysis

Based on existing research this paper focuses on σ-convergence and β-convergence analyses to calculate the convergence process of multidimensional relative poverty in China's ethnic areas, and also to be able to examine whether there is a catching-up effect between provinces at the spatial level.

σ-convergence refers to the declining trend of the multidimensional relative poverty level deviation between different regions as time advances. The coefficient of variation of the poverty level is used to determine whether there is convergence in the poverty index; if the coefficient of variation is shrinking year by year, it indicates that the difference in the poverty level between different regions is gradually declining, and at this time, there is σ-convergence. In this paper, the coefficient of variation is used to portray σ-convergence, and the model is as follows:

$$\sigma = \sqrt{\frac{\sum_{i=1}^{n} (D_{i,t} - \bar{D_{i,t}})^2}{n}} \Bigg/ \bar{D_{i,t}} \qquad (5)$$

where i denotes different provinces and t denotes the year, and $D_{i,t}$ is used to denote the multidimensional relative poverty index of province i at year t.

The theory of β-convergence reveals that regions with high levels of multidimensional relative poverty will catch up with regions with low levels of multidimensional relative poverty at a faster rate of development, thus gradually narrowing the differences between regions. Eventually, both high and low poverty areas will reach the same level of growth. This convergence phenomenon mainly includes two forms: absolute β-convergence and conditional β-convergence. Absolute β-convergence indicates that the multidimensional relative poverty levels of all provinces will converge to an identical steady-state level. This means that regardless of the

initial poverty level, all regions will reduce poverty at the same rate and gradually reach the same poverty level. Conditional β-convergence, on the other hand, means that controlling for the factors affecting the multidimensional relative poverty levels, the poverty levels of the provinces will show a tendency to converge. This means that despite the differences in poverty levels across provinces, they will all move towards similar steady-state levels.

Representing the multidimensional relative poverty of province i in year t by $D_{i,t}$, and the multidimensional relative poverty of province i in year t+1 by $D_{i,t+1}$, with $\mu_i$, $v_t$, and $\varepsilon_{it}$ denoting the spatial fixed effects, the time fixed effects, and the random perturbation term, respectively, the computational models for absolute β-convergence and conditional β-convergence are as follows:

$$\ln\left(\frac{D_{i,t+1}}{D_{i,t}}\right) = \alpha + \beta \ln D_{i,t} + \mu_i + v_t + \varepsilon_{it} \tag{6}$$

$$\ln\left(\frac{D_{i,t+1}}{D_{i,t}}\right) = \alpha + \beta \ln D_{i,t} + \lambda \sum_{j=1}^{n} Control_{i,t} + \mu_i + v_t + \varepsilon_{it} \tag{7}$$

α is the constant term, β is the convergence coefficient, if β is significantly less than 0, it indicates that there are convergence characteristics of multidimensional relative poverty, and vice versa, there are dispersion characteristics. The speed of convergence can be calculated by $-\ln(1+\beta)/T$. λ denotes the coefficient of the control variable and $Control_{i,t}$ is the control variable affecting the multidimensional relative poverty degree.

## Selection of indicators

Internationally, most of the indicators and dimensions are selected using the MPI method, which constructs the indicator system from micro-level indicators and then calculates them. However, considering China's actual national conditions, setting multidimensional relative poverty standards in China, especially in China's ethnic areas, does not have to be exactly the same as the international standards, instead, it should be localized and specific. In this paper, when setting the indicators, not only the economic dimension that reflects "poor", but also the social development dimension that can reflect the "difficulty", in addition to the ecological environment-related indicators are also included.

Based on Xiaolin Wang's indicator selection [2], we combine China's targeted policies implemented for ethnic regions and the geographic location characteristics of these areas. Furthermore, the following dimensional and indicator layers have been formulated for the next stage of development goals after achieving a moderately prosperous society in all aspects.

In the economic dimension, both income and employment are considered. Increasing income and promoting employment are important ways to alleviate relative poverty. On the income side, the methodology of a certain percentage of disposable income in OECD countries is drawn upon; on the employment side, the employment rate is used as a measurement indicator.

In terms of social development, factors such as consumption, education, healthcare, social security, and access to information are taken into account. The consumption behavior of residents has a direct impact on the overall economic demand and gross regional product. Additionally, it also influences the economic structure and industrial development. Therefore, the indicators of the consumption layer are selected as "per capita consumption expenditure" and "public service expenditures". With the rapid development of society and the economy, the demand for high-level talent in all walks of life is also growing, and the development of

postgraduate education is of great significance in improving the quality and level of higher education and promoting the progress and development of the region, so "the number of postgraduates enrolled in schools" is chosen as an indicator of the education level.

In the ecological environment dimension, taking into account the new development concept, the practical experience of ecological compensation for poverty eradication, and the people's demand for a better ecological environment, ecological environment-related indicators need to be included in the multidimensional poverty framework [28]. Water is an important resource for human life, socio-economic development, and ecosystem health [29]. Water resources have become an important factor restricting regional development and ecological maintenance [30]. For the human environment, two indicators, "gas penetration rate" and "daily water consumption", were selected, and for the ecological environment, "green space per capita" and "green coverage rate of built-up areas" were set. For the ecological environment, two indicators were set: "green space per capita" and "green coverage of built-up areas".

## Multidimensional relative poverty indicator system

When using indicator equal weighting, there are 15 indicators in 3 dimensions in multidimensional poverty measurement, and the weight of each indicator is 1/15. If we use the dimensional equal weighting method, the weight of each dimension is 1/3, and if there are n indicators in one of the dimensions, the weight of each indicator in the dimension is 1⁄3n. The multidimensional poverty incidence rate measured by the indicator equal weighting method is usually larger than that of the dimensional equal weighting method, so this paper adopts the dimensional equal weighting method, and the weight of each indicator is 1/3, and the specific weight of each indicator is in accordance with the dimensional equal weighting method. Therefore, this paper adopts the dimensional equal weight method, then the weight of each dimension of economy, social development, and ecological environment is 1/3, and the specific weight of each indicator is calculated according to the number of indicators in the dimension.

The poverty indicators selected in this paper are all positive, and based on the research results of Fengqin Wei, the poverty line (i.e., deprivation threshold) is set by the proportional method, and a certain proportion of the mean value of each indicator in 2021 is taken as the critical value of the indicator, and those lower than this threshold are deprived of the deprivation, and are assigned the value of 1, or else the value is 0 [31]. The multidimensional relative poverty indicator system constructed as a result is shown in Table 1 below:

## Data sources

The research object is the 12 provinces in China's ethnic areas, and the data are obtained from the China Statistical Yearbook, China Environmental Statistical Yearbook, and China Population and Employment Statistical Yearbook for the years 2012–2021, and supplemented with relevant statistics from the official websites of the provincial statistical bureaus, and missing data are made up by using the difference method.

## Empirical analysis

### Results of the relative poverty index measure

This paper calculates the deprivation status of the full sample across 15 indicator strata in three dimensions. The incidence of a single indicator reflects the deprivation of that indicator, and the contribution of an indicator implies the impact of that indicator on multidimensional relative poverty. Table 2 demonstrates the incidence of poverty indicators in each province. As a

**Table 1. Multidimensional relative poverty indicator system.**

| Dimension | Factor | Indicator | Poverty Line | Weight |
|---|---|---|---|---|
| Economic | income | per capita disposable personal income | 80% | 1/9 |
| | | fiscal budget revenue | 80% | 1/9 |
| | employment | employment rate | 99% | 1/9 |
| Social development | consume | per capita consumption expenditure of residents | 80% | 1/24 |
| | | public service expenditure | 80% | 1/24 |
| | education | educational fund | 80% | 1/24 |
| | | number of graduate students on campus | 70% | 1/24 |
| | healthcare | healthcare professionals per thousand population | 90% | 1/24 |
| | | average number of beds per thousand healthcare institutions | 90% | 1/24 |
| | facilities | bus ownership per 10000 people | 90% | 1/24 |
| | | number of urban lighting fixtures | 80% | 1/24 |
| Ecological | living | gas penetration rate | 90% | 1/12 |
| | | per capita daily water consumption | 80% | 1/12 |
| | natural | per capita park green space area | 80% | 1/12 |
| | | green coverage rate in built-up areas | 80% | 1/12 |

whole, the incidence of poverty in each province shows a downward trend, and the deprivation of the indicators has improved, with the largest decreases in Heilongjiang (from 10 to 1) and Gansu (from 12 to 3), the largest incidence of poverty in Tibet, and the smallest in Shaanxi.

Referring to the standard of the United Nations in measuring multidimensional poverty, this paper mainly studies and analyzes the situation when K = 0.3. Table 3 below shows the multidimensional relative poverty index of each province in ethnic areas obtained when K = 0.3. According to the assessment results of the multidimensional relative poverty index, we can see that the poverty situation in China's ethnic areas has shown a significant improvement trend in the past decade. During this period, the multidimensional relative poverty indexes of all provinces have shown a decreasing trend, with Heilongjiang province showing the most significant decrease, indicating that the province has made important progress in poverty reduction.

Cross-sectional comparisons show that the multidimensional relative poverty index of Tibet Province has been at a high level, indicating that the poverty situation in the province is still relatively serious and that it requires great attention and focus. In order to improve the poverty situation in Tibet Province, it should continue to persist in implementing effective

**Table 2. Incidence of poverty indicators.**

| | Inner Mongolia | Liao Ning | Ji Lin | Hei Long Jiang | Guang Xi | Yun Nan | Tibet | Shaan Xi | Gan Su | Qing Hai | Ning Xia | Xin Jiang |
|---|---|---|---|---|---|---|---|---|---|---|---|---|
| 2012 | 8 | 6 | 9 | 10 | 8 | 9 | 13 | 5 | 12 | 10 | 10 | 9 |
| 2013 | 8 | 6 | 10 | 7 | 6 | 9 | 13 | 5 | 11 | 11 | 9 | 8 |
| 2014 | 8 | 4 | 8 | 7 | 7 | 9 | 12 | 4 | 10 | 11 | 9 | 6 |
| 2015 | 6 | 3 | 7 | 7 | 7 | 8 | 12 | 4 | 12 | 11 | 9 | 5 |
| 2016 | 4 | 2 | 7 | 7 | 6 | 8 | 12 | 3 | 12 | 11 | 9 | 4 |
| 2017 | 4 | 1 | 8 | 6 | 5 | 9 | 11 | 2 | 9 | 9 | 9 | 4 |
| 2018 | 4 | 1 | 5 | 2 | 5 | 6 | 12 | 1 | 7 | 8 | 6 | 3 |
| 2019 | 2 | 1 | 5 | 2 | 3 | 3 | 12 | 1 | 5 | 5 | 6 | 2 |
| 2020 | 2 | 1 | 3 | 2 | 2 | 1 | 11 | 0 | 4 | 5 | 6 | 2 |
| 2021 | 2 | 0 | 3 | 1 | 1 | 1 | 10 | 0 | 3 | 5 | 6 | 1 |

**Table 3. Multidimensional relative poverty index.**

|  | 2012 | 2013 | 2014 | 2015 | 2016 | 2017 | 2018 | 2019 | 2020 | 2021 | total |
|---|---|---|---|---|---|---|---|---|---|---|---|
| Inner Mongolia | 0.444 | 0.444 | 0.444 | 0.292 | 0.208 | 0.208 | 0.208 | 0.125 | 0.125 | 0.125 | 0.133 |
| Liaoning | 0.403 | 0.403 | 0.250 | 0.167 | 0.125 | 0.042 | 0.042 | 0.153 | 0.153 | 0.111 | 0.081 |
| Jilin | 0.597 | 0.681 | 0.444 | 0.403 | 0.403 | 0.486 | 0.250 | 0.319 | 0.236 | 0.236 | 0.333 |
| Heilongjiang | 0.792 | 0.514 | 0.514 | 0.583 | 0.583 | 0.514 | 0.125 | 0.125 | 0.194 | 0.083 | 0.350 |
| Guangxi | 0.514 | 0.361 | 0.403 | 0.403 | 0.361 | 0.278 | 0.278 | 0.125 | 0.083 | 0.042 | 0.204 |
| Yunnan | 0.569 | 0.569 | 0.569 | 0.486 | 0.528 | 0.569 | 0.361 | 0.236 | 0.083 | 0.083 | 0.365 |
| Tibet | 0.806 | 0.806 | 0.722 | 0.722 | 0.722 | 0.681 | 0.722 | 0.722 | 0.639 | 0.528 | 0.707 |
| Shaanxi | 0.319 | 0.319 | 0.236 | 0.236 | 0.194 | 0.153 | 0.083 | 0.083 | 0.000 | 0.000 | 0.064 |
| Gansu | 0.764 | 0.681 | 0.639 | 0.722 | 0.722 | 0.556 | 0.431 | 0.347 | 0.306 | 0.264 | 0.517 |
| Qinghai | 0.597 | 0.681 | 0.681 | 0.681 | 0.681 | 0.556 | 0.514 | 0.278 | 0.278 | 0.278 | 0.439 |
| Ningxia | 0.708 | 0.514 | 0.514 | 0.514 | 0.514 | 0.514 | 0.319 | 0.319 | 0.319 | 0.319 | 0.456 |
| Xinjiang | 0.556 | 0.514 | 0.361 | 0.319 | 0.236 | 0.236 | 0.194 | 0.083 | 0.083 | 0.042 | 0.175 |

improvement measures and utilize its special advantages, such as tourism, to attract more business investment and residents' consumption, to promote the province's economic development. It is worth noting that the fact that Shaanxi Province's poverty index is equal to zero in 2020 and 2021 does not mean that Shaanxi Province has eliminated multidimensional relative poverty, but rather that it is performing well in comparison within the ethnic regions, but if one expands one's view to the whole of China, it is still lagging and there is still room for further improvement and development.

Based on the specific circumstances of each indicator layer being below the critical value in each year, the results of the multidimensional relative poverty index of China's ethnic areas decomposed according to dimensions in each year were calculated, as shown in Table 4. Since 2012, the multidimensional relative poverty reduction effect has been remarkable. The multidimensional relative poverty index declined from 0.589 in 2012 to 0.071 in 2021, showing a continuous downward trend and the largest decline in 2018.

In terms of the absolute changes in the poverty indexes of each dimension, the multidimensional poverty indexes of China's ethnic regions have shown a significant downward trend during the period under examination. This remarkable result shows the effectiveness of poverty reduction efforts in China's ethnic regions. It is worth noting that the contribution of the ecological dimension shows a downward trend, especially between 2019 and 2020 when the

**Table 4. Poverty index table decomposed by dimensions.**

| year | index | income | | social development | | ecological | |
|---|---|---|---|---|---|---|---|
|  |  | index | rate | index | rate | index | rate |
| 2012 | 0.589 | 0.204 | 34.6% | 0.233 | 39.5% | 0.153 | 25.9% |
| 2013 | 0.541 | 0.176 | 32.5% | 0.226 | 41.8% | 0.139 | 25.7% |
| 2014 | 0.441 | 0.139 | 31.5% | 0.205 | 46.5% | 0.097 | 22.0% |
| 2015 | 0.403 | 0.139 | 34.5% | 0.181 | 44.8% | 0.083 | 20.7% |
| 2016 | 0.376 | 0.130 | 34.5% | 0.163 | 43.4% | 0.083 | 22.2% |
| 2017 | 0.323 | 0.111 | 34.4% | 0.135 | 41.9% | 0.077 | 23.7% |
| 2018 | 0.196 | 0.074 | 37.9% | 0.094 | 47.9% | 0.028 | 14.2% |
| 2019 | 0.142 | 0.055 | 39.0% | 0.066 | 46.3% | 0.021 | 14.2% |
| 2020 | 0.105 | 0.046 | 44.0% | 0.052 | 49.5% | 0.007 | 6.6% |
| 2021 | 0.071 | 0.019 | 26.2% | 0.045 | 63.9% | 0.007 | 9.8% |

decline is most obvious. This is consistent with China's strategic direction of sustainable development, and also highlights China's determination and ability in ecological environmental protection and residential environment management. Although the poverty indices of the economic dimension and the social development dimension have declined, their contribution has gradually increased. This means that, in the current multidimensional poverty situation, economic and social development issues may become the most important issues to be concerned about in each region, and solving problems in the economic and social development dimensions should be the focus of current efforts.

Broken down to the level of indicators, on the economic side, the contribution of personal disposable income reaches 19.2%, which is the poverty indicator that accounts for the largest share of the 15 indicators, indicating that the income side is still an issue that the country needs to pay attention to in the long term. This is also in line with the 2019 proposal that "There are 600 million people with low and middle incomes in China and below, and their average monthly income is only about 1000 yuan." With the eradication of absolute poverty, China should also focus on raising the income of the population and emphasize the importance of income in poverty reduction. In respect of social development, the health contribution rate is the highest. On the one hand, medical and health resources in these areas are relatively scarce, with a relative lack of medical equipment and personnel, and in some areas, there are even no hospitals or medical institutions, making it impossible for residents to have easy access to medical services. On the other hand, due to various factors such as the geographical environment and ethnic culture, there is a need to improve the level of medical care and quality of service in ethnic minority areas. Additionally, medical and nursing staff in these areas have limited training opportunities, which makes it difficult for them to keep up with the pace of medical development. It is therefore necessary to increase investment in medical resources to promote the development of health in ethnic areas. Multidimensional relative poverty is a complex and multifaceted concept, involving economic, social, health, educational, environmental, and other factors. Cooperation and exchanges among provinces are also an important way to promote poverty reduction; sharing experiences and best practices can promote mutual learning and progress, thereby accelerating the process of poverty reduction in China's ethnic areas.

## Characteristics of temporal changes

Fig 1 depicts the trend of the multidimensional relative poverty index in ethnic areas over a period of 10 years based on different critical values of K. As the critical value of K increases, the multidimensional relative poverty index displays a decline. Furthermore, for the same critical value, the poverty index in each year is significantly lower than the previous year's score.

## Dagum Gini coefficient analysis

In dealing with multidimensional poverty, the threshold and dimension setting in the A-F method are of great significance in measuring poverty. However, the method considers those exceeding the threshold as non-poor samples and vice versa as poor samples, and such an assignment makes the calculated multidimensional relative poverty index lack coherence. In order to gain a deeper understanding of the differences in China's multidimensional relative poverty in different ethnic regions and its changes over time, this paper adopts the entropy method to provide a comprehensive evaluation of the multidimensional relative poverty situation in each province in China. Meanwhile, we calculated and decomposed the multidimensional overall Gini coefficient of relative poverty for ethnic areas in China from 2012 to 2021 using the Dagum Gini coefficient and its decomposition by subgroups. The Gini coefficient is

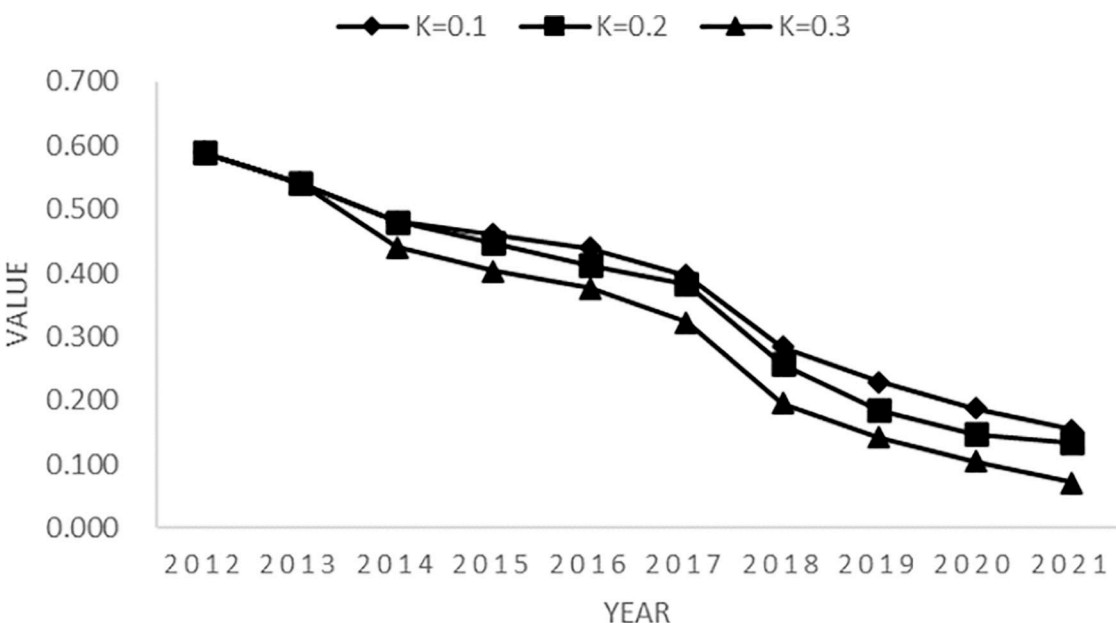

**Fig 1. Multidimensional relative poverty index at different values of K.**

an important indicator of inequality in income or consumption within a country or region. In this context, it can be used to measure the degree of inequality in the distribution of multidimensional relative poverty. Through the entropy method, the weights of different dimensions can be taken into account to reflect the status of multidimensional poverty more comprehensively. This method allows for a comprehensive evaluation of the multidimensional poverty situation in each province and a comparison of the differences between different years, thus providing insight into the geographical differences and dynamic evolution of multidimensional relative poverty in China.

The weights of the indicators calculated by the entropy value method are shown in Table 5, in which "number of graduate students on campus" has the largest weight, accounting for 19.1594%, and "the coverage rate of greening area in built-up areas" has the lowest weight, accounting for 0.9871%.

The Dagum Gini coefficient is a measurement for research on the problem of disequilibrium. First: the overall Dagum Gini coefficient = intra-group Gini coefficient Gw + intergroup Gini coefficient Gb + hypervariable density Gini coefficient Gt; second: the contribution rate is the proportion of intra-group Gini coefficient Gw, inter-group Gini coefficient Gb or

**Table 5. Weights of indicators calculated by entropy method.**

| indicator | weight | indicator | weight |
|---|---|---|---|
| per capita disposable personal income | 0.05946 | average number of beds per thousand healthcare institutions | 0.029799 |
| fiscal budget revenue | 0.10860 | | |
| employment rate | 0.05444 | bus ownership per 10000 people | 0.036131 |
| per capita consumption expenditure of residents | 0.03996 | per capita daily water consumption | 0.11692 |
| public service expenditure | 0.07639 | gas penetration rate | 0.013319 |
| number of graduate students on campus | 0.19159 | per capita park green space area | 0.035977 |
| educational fund | 0.08553 | number of urban lighting fixtures | 0.106083 |
| healthcare professionals per thousand population | 0.035905 | green coverage rate in built-up areas | 0.009871 |

**Table 6. Dagum Gini coefficient and contribution rate.**

| year | Gini coefficient | | | | contribution rate | | |
|------|-------|-------|-------|-------|-------------|-------------|---------------------|
| | total | Gw | Gb | Gt | within-group | intergroup | hypervariable density |
| 2012 | 0.197 | 0.042 | 0.118 | 0.037 | 21.090% | 59.894% | 19.016% |
| 2013 | 0.182 | 0.040 | 0.108 | 0.035 | 21.680% | 59.153% | 19.167% |
| 2014 | 0.175 | 0.039 | 0.100 | 0.036 | 22.443% | 57.134% | 20.423% |
| 2015 | 0.175 | 0.045 | 0.079 | 0.052 | 25.366% | 44.799% | 29.835% |
| 2016 | 0.177 | 0.043 | 0.080 | 0.054 | 24.024% | 45.272% | 30.704% |
| 2017 | 0.186 | 0.048 | 0.058 | 0.079 | 26.019% | 31.394% | 42.587% |
| 2018 | 0.188 | 0.043 | 0.080 | 0.065 | 22.749% | 42.705% | 34.547% |
| 2019 | 0.177 | 0.043 | 0.073 | 0.061 | 24.138% | 41.187% | 34.676% |
| 2020 | 0.167 | 0.039 | 0.078 | 0.049 | 23.405% | 47.063% | 29.531% |
| 2021 | 0.157 | 0.036 | 0.082 | 0.039 | 23.233% | 51.996% | 24.771% |

hypervariable density Gini coefficient Gt. Table 5 provides the weights for each indicator. Using these weights, the composite score for each indicator can be calculated. The Dagum Gini coefficient is then measured and analyzed to obtain the coefficient value and contribution rate of each part.

Table 6 demonstrates the overall Gini coefficient and the contribution of each component of multidimensional relative poverty in ethnic areas. The overall Gini coefficient shows a decreasing trend, with a slight increase in the three years from 2016 to 2018. The decrease in the value of the Gini coefficient indicates that the multidimensional relative poverty gap in ethnic areas is reducing. In terms of evolution, the largest decrease was observed from 2012 to 2013, but the decreasing trend was not significant in the following years and there was an upward trend, which began to decrease year by year after 2018 and dropped to a minimum value of 0.157 in 2021.

In terms of dynamic changes, the changes in the contribution of intra-group disparity, inter-group disparity, and hypervariable density to the total disparity are different. Regarding the contribution rate of intra-regional disparity, its change has been relatively smooth, hovering between 21% and 26%, with a relatively small change. Both the intra-group contribution rate and the hypervariable density contribution rate have shown a trend of increasing and then decreasing, and the hypervariable density contribution rate has moved just the opposite of the inter-group disparity contribution rate, which reached a maximum of 42.587% in 2017, the only year in which it exceeded the inter-group contribution rate in a province. Analyzing the level of contribution, the intergroup disparity has the highest contribution to the overall disparity, and was the main source of the overall disparity for 10 years, reaching a maximum of 59.894%, with an average annual contribution rate of 48.06%, a value higher than that of the other two components in any given year. In this decade, the intra-group contribution turned out to be the component with the lowest share, while the inter-group contribution declined but still accounted for a larger share. This shows that the overall differences in ethnic areas are decreasing, but they are still the main source of regional differences, which points to the need to focus on the reduction of inter-group differences to effectively combat relative poverty and reduce the differences between ethnic areas.

The within-group Gini coefficient decomposition shows the Gini coefficient data of each group: firstly, to analyze the trend of the Gini coefficient of each group as a whole; secondly, to comprehensively compare the size of the Gini coefficient of each group. **Fig 2** is the decomposition of the Gini coefficient within the group, because in the selected samples, there is only one province in North China, Inner Mongolia, and only one province in South China,

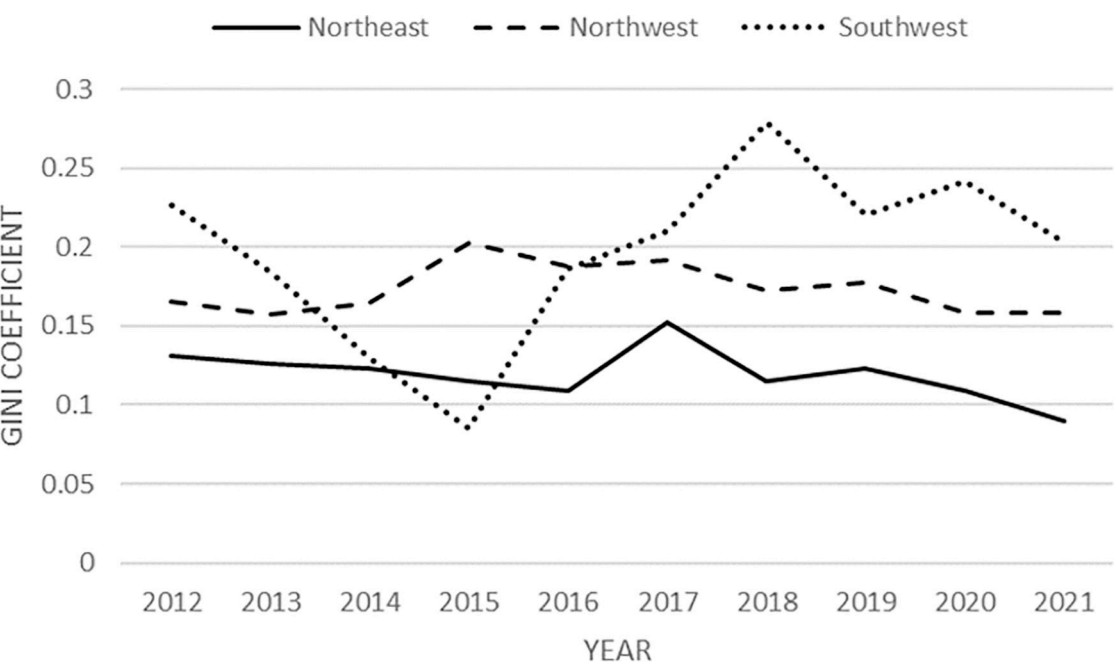

**Fig 2. The Gini coefficient within the group.**

Guangxi, so the Gini coefficient of these two regions corresponding to the group of 0, are overlapped with the x-axis and are not displayed in the following figure.

As can be seen from **Fig 2**, the Southwest region (Tibet and Yunnan Provinces) has the biggest internal differences, and its intra-group Gini coefficient maintains a high level and is similar to the trend of the overall Gini coefficient. Although Tibet and Yunnan share a border, they have significant differences in their climates, topography, and industrial development. These differences have led to greater variation in poverty levels within them. The intra-group Gini coefficients of the Northwest and Northeast regions have changed less over the 10 years, and, except for a few individual years, they have been on a relatively steady downward trend. In addition, the intra-group Gini coefficient of the Northeast region is the smallest (except for 2015), and the three provinces in the Northeast region are geographically similar, with not much difference in their development patterns and levels of development, so the internal differences in the degree of poverty are also smaller.

## Convergence analysis

After the spatial and temporal distribution of multidimensional poverty and the decomposition of differences, to further understand the evolutionary trend of the differences in multidimensional relative poverty in ethnic areas, this paper utilizes the σ-convergence model and the β-convergence model to verify its convergence.

**σ-convergence test.** **Fig 3** illustrates the evolutionary trend of the coefficient of variation of multidimensional relative poverty in ethnic areas. As can be seen from the figure, the coefficient of variation in ethnic areas exhibited a downward trend, but the fluctuation was small. Overall, the coefficient of variation of ethnic regions changed from 0.484 to 0.313 during the sample period, the imbalance of spatial differences gradually weakened, and the synergistic effect between regions increased. The coefficient of variation increased in individual years after 2015, and in addition, the convergence phenomenon was not significant. The coefficient

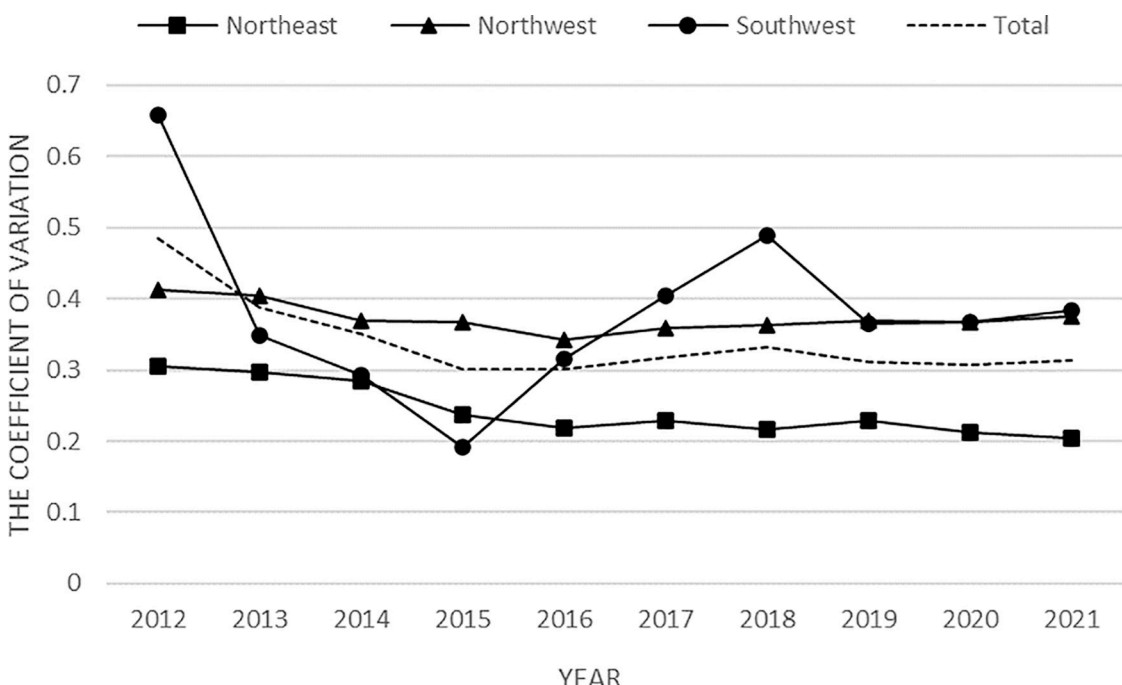

**Fig 3. The evolutionary trend of the coefficient of variation.**

of variation in the Southwest region shows a "falling-rising-falling" trend, with a large fluctuation amplitude. There was a significant increase in 2015, and no overall σ-convergence. The coefficients of variation in the Northwest and Northeast regions both show a general trend of decline, but the decline in the Northwest region is smaller, so there is no σ-convergence feature on the whole. The Northeast region has a larger decline, and the regional differences are gradually narrowing, and there is a σ-convergence feature on the whole, which is consistent with the conclusions drawn from the Dagum Gini coefficient analysis mentioned above.

**Absolute β-convergence.**   Table 7 displays the status of absolute β-convergence of multidimensional relative poverty in ethnic regions. As can be seen from the Table 7, there is an absolute β-convergence in all regions, except for the Northeast region. This means that without considering other variables such as economic development and ecological environment, regions with high multidimensional poverty index will have a faster development speed and

**Table 7. The result of absolute β-convergence.**

| variable | ethnic area | northeast | southwest | northwest |
|---|---|---|---|---|
| convergence coefficient | -0.386*** (0411) | 0.022 (0.0539) | -0.556*** (0.0955) | -0.078** (0.0319) |
| constant term | -0.102*** (0.0208) | 0.049*** (0.0128) | -0.274*** (0.0782) | 0.018 (0.0185) |
| area fixed effects | yes | yes | yes | yes |
| time fixed effects | yes | yes | yes | yes |
| convergence rate | 0.048 | | 0.081 | 0.008 |
| $R^2$ | 0.2399 | 0.0074 | 0.4537 | 0.1335 |
| F | 5.28*** | 0.62 | 12.54*** | 2.09** |

Note

*, **, *** denote significance at the 10%, 5%, and 1% levels, respectively, with standard errors in parentheses, below.

**Table 8. The result of conditional β-convergence.**

| variable | ethnic area | northeast | southwest | northwest |
|---|---|---|---|---|
| convergence coefficient | -0.8844*** (0.0440) | -0.2274* (0.1242) | -0.9085*** (0.0898) | -0.4325*** (0.1316) |
| per capita consumption expenditure of residents | 0.00001*** (0.00000) | 0.0000** (0.0000) | 0.0000 (0.00005) | 0.00003*** (0.00001) |
| healthcare professionals per thousand population | 0.0714*** (0.0146) | 0.0239 (0.0152) | 0.1242 (0.1297) | -0.0157 (0.0187) |
| per capita park green space area | 0.0203*** (0.0060) | -0.0229* (0.0114) | 0.0450** (0.0180) | 0.0152* (0.0078) |
| constant term | -1.3079***(0.1034) | -0.0287 (0.1723) | -1.6239*** (0.2507) | -0.3825 (0.2285) |
| $R^2$ | 0.1818 | 0.3001 | 0.5299 | 0.0036 |
| F | 31.72*** | 0.88 | 27.10*** | 5.47*** |

can catch up with regions with low multidimensional poverty index by utilizing the "catching-up effect". As a result, the multidimensional relative poverty of each region will converge to the same steady state level over time. The absolute β-convergence coefficient of the multidimensional relative poverty of the ethnic regions as a whole, as well as the Southwest region and the Northwest region, are all significantly negative at least at the 5% level. This indicates that the regions with higher multidimensional poverty indexes will have a faster development speed. In addition, the rate of convergence of multidimensional relative poverty in the ethnic regions is 0.048, while that in the southwestern and northwestern regions is 0.081 and 0.008, respectively. Therefore, it can be concluded that the multidimensional poverty in the southwestern region has the fastest rate of convergence.

**Conditional β-convergence.** Table 8 shows the specific results of using the formula for conditional convergence to calculate the coefficient of conditional β-convergence of multidimensional relative poverty from 2012 to 2021. The regression results indicate that there is a significant negative coefficient of conditional β-convergence in the ethnic regions, the northeast, the southwest, and the northwest. This means that over time, the multidimensional relative poverty in each province will converge to their respective steady-state levels. The Southwest region has the fastest speed of convergence.

To sum up, the reason for the overall absolute β-convergence in ethnic regions may lie in the fact that China strongly supports industrial development and rural revitalization in ethnic backward regions, and has introduced a series of policies to promote the economic development, social development, and ecological environment of ethnic regions. The support and inclination of the policies obviously make the multidimensional relative poverty situation in the backward areas improve at a faster rate, thus making the overall multidimensional relative poverty show absolute β-convergence. As for conditional β-convergence, the absolute values of the conditional β-convergence coefficients for the ethnic regions as a whole and each region are greater than the absolute values of their absolute β-convergence coefficients, which reflects the fact that control variables such as per-capita consumption expenditures, health technicians per 1,000 population, and per-capita green space in parks have a certain contributing effect on the convergence of the ethnic regions as a whole as well as the convergence of the multidimensional relative poverty in each region. In addition, for the Northeast region, the absence of absolute β-convergence but the presence of conditional β-convergence may be due to the fact that the per capita consumption expenditures of Heilongjiang and Jilin Provinces are relatively low, which leads to a widening of the gap between the three provinces in the Northeast region.

## Discussion

After solving the problem of absolute poverty, the governance of relative poverty has become a new challenge faced by ethnic areas. Research has shown that the overall poverty situation in

ethnic regions has improved, but there is a trend towards imbalanced and dispersed poverty. Xizang's poverty index has been at the highest level in the study year. As the level of deprivation K increases, the incidence of multidimensional poverty and the multidimensional poverty index decrease, and the average share of deprivation increases. Most ethnic regions are located on the border of China, with relatively lagging economic development and a large number of impoverished people. Especially in the western region, due to its harsh ecological environment, lack of natural resources, and fragile industrial foundation, a series of problems have led to complex poverty factors. Despite ensuring basic survival conditions, these regions still face multidimensional poverty issues such as health, culture, and education.

The governance of relative poverty is a long-term issue that requires consideration and attention. There are certain shortcomings in this study, and it is believed that the field of multidimensional relative poverty in the future can be further explored and enriched in the following aspects:

In terms of the research area, this article collects data from the provincial level to conduct research, and there may be significant disparities in poverty within a province. This requires further optimization of the selection of indicators and thresholds in this study. In order to improve the accuracy and completeness of the multidimensional relative poverty measurement system, future research can subdivide the research objects. Determine an appropriate evaluation system based on the characteristics of the research objects, in order to obtain a more accurate poverty index and improve the accuracy of multidimensional relative poverty measurement.

In terms of the research groups, this study analyzed the multidimensional relative poverty levels and dynamic evolution in ethnic regions but did not involve research on special groups such as the elderly, disabled, and mobile populations. Because the living conditions and goals of these groups differ significantly from those of the general population, scholars can strengthen their research on the multidimensional relative poverty issues of special groups. Set identification standards based on their characteristics to develop appropriate solutions.

## Conclusion and recommendations

Most existing studies on multidimensional poverty adopt an absolute perspective and focus on the entire country. This method does not take into account the development gap and characteristic differences between regions, and is not suitable for studying relative poverty in ethnic regions. In response to this deficiency, this article considers the internal characteristics of ethnic regions and calculates the multidimensional relative poverty index from a multidimensional perspective. Through further analysis, regional disparities and convergence changes were analyzed based on the poverty index. In terms of theory, a more comprehensive measurement system has been formed, providing reference for future research on ethnic regions. In terms of practical significance, it has guiding significance for the country to formulate development strategies for ethnic regions.

### Conclusion

In this paper, we constructed a multidimensional relative poverty index system and used the entropy method to conduct a comprehensive evaluation, used the obtained comprehensive score to calculate the Dagum Gini coefficient to analyze its spatiotemporal differences, and finally used the σ-convergence model and the β-convergence model to test the convergence of the multidimensional relative poverty in the ethnic areas, and the results were found:

1. The multidimensional relative poverty indexes of all provinces in the ethnic regions show a decreasing trend, in which the contribution rate of the ecological environment dimension

is decreasing, while the contribution rate of the social development dimension shows an increasing trend.

2. The overall, intra-group and inter-group gaps in the ethnic regions are decreasing, and there is a significant difference in the multidimensional relative poverty among the provinces, with the multidimensional poverty in the southwest region always at the highest level, and the multidimensional poverty in the northeast region is the lowest compared to that in the northeast region.

3. The convergence results show that there is no σ-convergence of multidimensional relative poverty in ethnic regions, but there is absolute β-convergence and conditional β-convergence, and in terms of subregions, absolute β-convergence, and conditional β-convergence exist in the Southwest region and the Northwest region, while there is no absolute β-convergence in the Northeast region, and the Southwest region has the fastest rate of convergence in multidimensional relative poverty.

## Recommendation

As an indispensable part of Chinese culture, ethnic culture contains the labor wisdom and unique values of each ethnic group. The relevant department can formulate policies conducive to the innovation and development of national culture, encourage more enterprises to enter the national culture enterprises and promote the upgrading and flourishing development of the culture industry, which will help realize the revitalization of the nationalities and the cultivation of talents, and provide more solid and abundant talents for the nationalities.

To realize the all-round development of economy, society, and ecology in ethnic areas, industrial support and government support are very important foundations. Only through industrial revitalization can we promote residents' consumption, increase employment opportunities, and improve people's living standards. In ethnic areas, the traditional economic structure is no longer adapted to the new development requirements, and there is an urgent need to carry out industrial reform and upgrading, and to promote the economic growth mode of ethnic areas to a more sustainable direction.

A good ecological environment is the greatest advantage of ethnic regions, which have unique advantages in terms of geography and natural endowments. The development of local tourism can be stimulated through the protection of the ecological environment and the development of green ecological resources, which will improve the level of economic development. Local governments should strengthen the comprehensive management of environmental problems and improve the institutional mechanism for environmental protection, to provide an ecological foundation for the realization of the development of ethnic regions.

## Supporting information

**S1 Data.**
(XLSX)

## Author Contributions

**Conceptualization:** Jing Cheng.

**Data curation:** Xiaobin Yu.

**Formal analysis:** Jing Cheng.

**Funding acquisition:** Jing Cheng.

**Investigation:** Jing Cheng.

**Methodology:** Jing Cheng.

**Project administration:** Jing Cheng.

**Resources:** Xiaobin Yu.

**Software:** Xiaobin Yu.

**Supervision:** Jing Cheng.

**Validation:** Jing Cheng.

**Visualization:** Jing Cheng.

**Writing – original draft:** Jing Cheng.

**Writing – review & editing:** Jing Cheng.

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
