## [Decision Letter · Decision Letter 0]

26 Feb 2024

PONE-D-24-03074Spatial and Temporal Differences and Convergence Analysis of Multidimensional Relative Poverty in Ethnic AreasPLOS ONE

Dear Dr. Xiao Bin,

Thank you for submitting your manuscript to PLOS ONE. After careful consideration, we feel that it has merit but does not fully meet PLOS ONE’s publication criteria as it currently stands. Therefore, we invite you to submit a revised version of the manuscript that addresses the points raised during the review process.

We look forward to receiving your revised manuscript.

Kind regards,

Dingde Xu

Academic Editor

PLOS ONE

Journal Requirements:

"This study was funded by: National Social Science Fund of China (Grant numbers 23BJY182); The Ministry of education of Humanities and Social Science project of China (Grant numbers 22YJA790010)"

Additional Editor Comments:

Now we have received valid comments from two reviewers, and based on their suggestions, I propose to give this article a minor revision decision.

Reviewers' comments:

Reviewer's Responses to Questions

**Comments to the Author**

1. Is the manuscript technically sound, and do the data support the conclusions?

Reviewer #1: Yes

Reviewer #2: Yes

2. Has the statistical analysis been performed appropriately and rigorously? 

Reviewer #1: Yes

Reviewer #2: Yes

3. Have the authors made all data underlying the findings in their manuscript fully available?

Reviewer #1: Yes

Reviewer #2: Yes

4. Is the manuscript presented in an intelligible fashion and written in standard English?

Reviewer #1: Yes

Reviewer #2: Yes

5. Review Comments to the Author

Reviewer #1: The paper can be accepted after making the following minor revisions.

(1) The innovation of this paper needs to be highlighted in the abstract.

(2) The author needs to introduce the reasons for the research method and combine it with the innovation of the research in this paper.

(3) This article has obtained some interesting findings through the models, but these findings need to be further verified from theory or actual conditions. Also, further highlight the contribution of this article.

(4) For readers to quickly catch your contribution, it would be better to highlight major difficulties and challenges, and your original achievements to overcome them, in a clearer way in the abstract and introduction.

(5) The article lacks an important discussion link, in which the author should focus on describing the differences between the article study and other scholars' studies, thus highlighting the relevance and academic value of the article, the following literature should be helpful for your research：(1) Reduction pathways identification of Agricultural Water Pollution in Hubei Province, China. (2) A Set Pair Analysis Method for Assessing and Forecasting Water Conflict Risk in Transboundary River Basins

(6) Compared with the available literature, what are the theoretical contributions and application values of this study? It is suggested to enhance the corresponding discussions in the conclusion part.

Reviewer #2: The current world economic development is unbalanced, with a large wealth gap, and reducing poverty has become an important issue facing the world. This article takes 12 ethnic regions in China as the research object, constructs a poverty measurement system using Gini coefficient and convergence analysis, evaluates the level of regional poverty, and provides suggestions, which has important theoretical and practical significance. The modification suggestions are as follows:

1.The citation format of the literature is incorrect and differs from the requirements of this journal. It is recommended to make corrections.

2.It is recommended to write the shortcomings and advantages of this research field in the preface, as well as the innovative points of this study.

3.The format after the formula is incorrect and should be (1) (2) The three line table is not up to standard.

4.Center the table title and place it above the table.

6. PLOS authors have the option to publish the peer review history of their article (what does this mean?). If published, this will include your full peer review and any attached files.

Reviewer #1: No

Reviewer #2: No

---

## [Author Response · Author response to Decision Letter 0]

7 Mar 2024

SUGGESTIONS FROM EDITOR

1.Please ensure that your manuscript meets PLOS ONE’s style requirements, including those for file naming.

Reply: The content of the article has been changed according to the style template, and the file name has also been modified according to the requirements.

2. Please state what role the funders took in the study. If the funders had no role, please state: “The funders had no role in study design, data collection and analysis, decision to publish, or preparation of the manuscript”. If this statement is not correct you must amend it as needed. Please include this amended Role of Funder statement in your cover letter; we will change the online submission form on your behalf.

Reply: The funders had no role in study design, data collection and analysis, decision to publish, or preparation of the manuscript.

3.Please provide a complete Data Availability Statement in the submission form, ensuring you include all necessary access information or a reason for why you are unable to make your data freely accessible. If your research concerns only data provided within your submission, please write “All data are in the manuscript and/or supporting information files” as your Data Availability Statement.

Reply: All data are in the manuscript and supporting information files. 

4.Please review your reference list to ensure that it is complete and correct. If you have cited papers that have been retracted, please include the rationale for doing so in the manuscript text, or remove these references and replace them with relevant current references. Any changes to the reference list should be mentioned in the rebuttal letter that accompanies your revised manuscript. If you need to cite a retracted article, indicate the article’s retracted status in the References list and also include a citation and full reference for the retraction notice.

Reply: There are no cases where the cited references have been retracted. Three new references have been added to the article, “Reduction pathways identification of Agricultural Water Pollution in Hubei Province, China” and “A Set Pair Analysis Method for Assessing and Forecasting Water Conflict Risk in Transboundary River Basins”. The two articles mention water as an important resource for human life, socio-economic development and ecosystem health. Water conflict has become an important factor restricting regional economic development and social stability, and accurate assessment of water conflict risk can timely grasp the water conflict situation in transboundary river basins and provide decision-making basis for risk prevention. This provides help for the per capita daily water consumption and ecological environment in the poverty measurement system in this paper. “The effect of digital infrastructure development on enterprise green transformation”. The green enterprises mentioned in this article also provide assistance in the construction of indicator systems.

Reviewer #1:

1.The innovation of this paper needs to be highlighted in the abstract.

Reply: The abstract section has added the innovative point of the article, which reads: “（1）Most of the existing research on poverty in China focuses on countries or individual provinces, and this paper focuses on ethnic minority areas that are often overlooked. （2）The measurement system of the existing studies is relatively simple, and most of the indicators are derived from international standards, which is not suitable for ethnic areas in China. This paper combines unique factors such as geographical characteristics and policies of ethnic minority areas to enrich the poverty measurement system.” A detailed description of the innovations is also described in the introduction.

2. The author needs to introduce the reasons for the research method and combine it with the innovation of the research in this paper.

Reply: The multidimensional poverty measurement method used in this paper is the double critical value method, which is widely popular in the world. The calculation method of the AF method is simple and easy to understand, and generally includes four parts: the value setting of each dimension and index, the weight setting of each dimension and index, the poverty identification of each dimension, and the sum and decomposition of poverty. Unlike most people, for the weight setting, this paper uses the entropy weight method to calculate the weight, which can avoid the interference of subjective factors.

3. This article has obtained some interesting findings through the models, but these findings need to be further verified from theory or actual conditions. Also, further highlight the contribution of this article.

Reply: This article constructs a multidimensional relative poverty indicator system, and uses the calculated comprehensive scores to analyze spatiotemporal differences and convergence. The following conclusions have been drawn:（1）The multidimensional relative poverty index of each province in ethnic regions is showing a downward trend, with the contribution rate of the ecological environment dimension decreasing and the contribution rate of the social development dimension showing an upward trend.（2）The multidimensional poverty situation in Xizang is the most serious.（3）The poverty situation in the Southwest region is more complex and severe than that in the Northeast region. These conclusions are consistent with the actual development status of ethnic regions in China. This study can provide more specific reference directions for formulating policies to alleviate multidimensional poverty, as well as dimensions and indicator directions for narrowing regional disparities. This also verifies the effectiveness of China's policy implementation in ethnic regions.

4.For readers to quickly catch your contribution, it would be better to highlight major difficulties and challenges, and your original achievements to overcome them, in a clearer way in the abstract and introduction.

Reply: The research difficulties on multidimensional relative poverty and the achievements of this study have been added in the introduction. The specific content is: “In the long run, eliminating difficulties such as relative poverty, relative backwardness, and regional disparities will be a major challenge. To solve the problem of relative poverty, the first thing to do is to develop accurate methods for identifying relative poverty and construct a reasonable multidimensional indicator system for relative poverty. In addition, as the indicator system changes, it will inevitably have an impact on identifying impoverished groups, so dynamic identification methods should be adopted to consolidate poverty alleviation achievements”.

5.The article lacks an important discussion link, in which the author should focus on describing the differences between the article study and other scholars’ studies, thus highlighting the relevance and academic value of the article, the following literature should be helpful for your research: (1) Reduction pathways identification of Agricultural Water Pollution in Hubei Province, China. (2) A Set Pair Analysis Method for Assessing and Forecasting Water Conflict Risk in Transboundary River Basins.

Reply: A discussion section has been added to the article. The specific content is “After solving the problem of absolute poverty, the governance of relative poverty has become a new challenge faced by ethnic areas. Research has shown that the overall poverty situation in ethnic regions has improved, but there is a trend towards imbalanced and dispersed poverty. Xizang's poverty index has been at the highest level in the study year. As the level of deprivation K increases, the incidence of multidimensional poverty and the multidimensional poverty index decrease, and the average share of deprivation increases. Most ethnic regions are located on the border of China, with relatively lagging economic development and a large number of impoverished people. Especially in the western region, due to its harsh ecological environment, lack of natural resources, and fragile industrial foundation, a series of problems have led to complex poverty factors. Despite ensuring basic survival conditions, these regions still face multidimensional poverty issues such as health, culture, and education.

The governance of relative poverty is a long-term issue that requires consideration and attention. There are certain shortcomings in this study, and it is believed that the field of multidimensional relative poverty in the future can be further explored and enriched in the following aspects:

In terms of the research area, this article collects data from the provincial level to conduct research, and there may be significant disparities in poverty within a province. This requires further optimization of the selection of indicators and thresholds in this study. In order to improve the accuracy and completeness of the multidimensional relative poverty measurement system, future research can subdivide the research objects. Determine an appropriate evaluation system based on the characteristics of the research objects, in order to obtain a more accurate poverty index and improve the accuracy of multidimensional relative poverty measurement.

In terms of the research groups, this study analyzed the multidimensional relative poverty levels and dynamic evolution in ethnic regions but did not involve research on special groups such as the elderly, disabled, and mobile populations. Because the living conditions and goals of these groups differ significantly from those of the general population, scholars can strengthen their research on the multidimensional relative poverty issues of special groups. Set identification standards based on their characteristics to develop appropriate solutions”. 

The two provided literature mention that “water conflict has become an important factor restricting regional economic development and social stability” and “existing research mostly focuses on static evaluation of past and present conflicts, and long-term prediction can take corresponding effective measures in advance to avoid risks or reduce the harm caused by conflicts”. These two points, as well as other writing content in the article, have helped me and have been cited in this article.

6.Compared with the available literature, what are the theoretical contributions and application values of this study? It is suggested to enhance the corresponding discussions in the conclusion part.

Reply: The theoretical contribution and application value of this article have been supplemented in the conclusion, with the specific content as follows: “Most existing studies on multidimensional poverty adopt an absolute perspective and focus on the entire country. This method does not take into account the development gap and characteristic differences between regions, and is not suitable for studying relative poverty in ethnic regions. In response to this deficiency, this article considers the internal characteristics of ethnic regions and calculates the multidimensional relative poverty index from a multidimensional perspective. Through further analysis, regional disparities and convergence changes were analyzed based on the poverty index. In terms of theory, a more comprehensive measurement system has been formed, providing reference for future research on ethnic regions. In terms of practical significance, it has guiding significance for the country to formulate development strategies for ethnic regions”.

Reviewer #2:

1.The citation format of the literature is incorrect and differs from the requirements of this journal. It is recommended to make corrections.

Reply: The literature format has been revised according to the requirements of the journal.

2.It is recommended to write the shortcomings and advantages of this research field in the preface, as well as the innovative points of this study.

Reply: The introduction has been supplemented with information on the shortcomings and advantages of existing research, as well as the innovative points of this article. The specific content is: “In summary, scholars have conducted extensive research on the field of multidimensional relative poverty. Significant research results have been achieved in both the theoretical analysis of multidimensional poverty and relative poverty, as well as the exploration of poverty influencing factors and policy formulation. However, through in-depth reading of the literature, it has been found that there may be several shortcomings in current research on multidimensional relative poverty. Firstly, most of the current research focuses on the overall country or individual provinces, which makes it difficult to identify poverty differences in different regions and also makes it difficult to see poverty associations between regions. Secondly, in terms of indicator selection, many studies use international standards and the poverty situation of developed countries as benchmarks, which may result in the selected indicators not being suitable for measuring the multidimensional relative poverty level in ethnic regions of China. On the premise of borrowing and learning from the research results of the former, there may be several innovations in this article: firstly, this article focuses on ethnic regions that are easily overlooked. Ethnic regions have complex and diverse geographical, cultural, and economic characteristics, with significant spatial differences in multidimensional poverty. This article uses the AF method to measure the multidimensional relative poverty level in ethnic areas, providing theoretical value for subsequent research on its influencing factors. Secondly, by combining the characteristics of ethnic regions and specific poverty situations, the multidimensional relative poverty measurement system has been enriched. This article selects evaluation indicators that are more in line with the actual situation of ethnic regions, and obtains a more accurate and practical poverty index through the constructed evaluation system. Thirdly, this article uses the Dagum Gini coefficient to analyze the spatiotemporal differences in multidimensional poverty in ethnic regions, taking into account the differences and sources of poverty between different regions”.

3.The format after the formula is incorrect and should be (1) (2). The three-line table is not up to standard.

Reply: The formula number has been changed to (1) (2). The format of the three-line table has been corrected according to the requirements of the journal.

4.Center the table title and place it above the table.

Reply: The table title has been centered and placed above the table.

---

## [Editor Report · Decision Letter 1]

20 Mar 2024

Spatial and temporal differences and convergence analysis of multidimensional relative poverty in ethnic areas

PONE-D-24-03074R1

Dear Dr. Yu,

We’re pleased to inform you that your manuscript has been judged scientifically suitable for publication and will be formally accepted for publication once it meets all outstanding technical requirements.

Kind regards,

Dingde Xu

Academic Editor

PLOS ONE

Additional Editor Comments (optional):

The authors have addressed well with the reviewers' comments and it is suggested to be accepted.
---

## [Editor Report · Acceptance letter]

25 Mar 2024

PONE-D-24-03074R1 

PLOS ONE

Dear Dr. Yu, 

I'm pleased to inform you that your manuscript has been deemed suitable for publication in PLOS ONE. Congratulations! Your manuscript is now being handed over to our production team.

Kind regards, 

on behalf of

Dr. Dingde Xu 

Academic Editor

PLOS ONE